# The Effect of Indole-3-Lactic Acid from *Lactiplantibacillus plantarum* ZJ316 on Human Intestinal Microbiota In Vitro

**DOI:** 10.3390/foods11203302

**Published:** 2022-10-21

**Authors:** Qingqing Zhou, Zuorui Xie, Danli Wu, Lingli Liu, Yongqing Shi, Ping Li, Qing Gu

**Affiliations:** Key Laboratory for Food Microbial Technology of Zhejiang Province, College of Food Science and Biotechnology, Zhejiang Gongshang University, Hangzhou 310018, China

**Keywords:** indole-3-lactic acid, *Lactiplantibacillus plantarum* ZJ316, antibacterial activity, intestinal microbiota, short-chain fatty acids

## Abstract

Microbiota-derived tryptophan metabolites are essential signals for maintaining gut homeostasis, yet the potential contribution to modulating gut microbiota has been rarely investigated. In this study, *Lactiplantibacillus plantarum* ZJ316 (CCTCC No. M 208077) with a high production (43.14 μg/mL) of indole-3-lactic acid (ILA) was screened. ILA with 99.00% purity was prepared by macroporous resin, Sephadex G–25 and reversed-phase high-performance liquid chromatography. Purified ILA can effectively inhibit foodborne pathogens such as *Salmonella* spp., *Staphylococcus* spp., *Escherichia coli* and *Listeria monocytogenes*. In an in vitro model of the human gut microbiota, a medium-dose ILA (172 mg/L) intervention increased the average relative abundance of phyla Firmicutes and Bacteroidota by 9.27% and 15.38%, respectively, while Proteobacteria decreased by 14.36% after 24 h fermentation. At the genus level, the relative abundance of *Bifidobacterium* and *Faecalibacterium* significantly increased to 5.36 ± 2.31% and 2.19 ± 0.77% (*p* < 0.01), respectively. *Escherichia* and *Phascolarctobacterium* decreased to 16.41 ± 4.81% (*p* < 0.05) and 2.84 ± 1.02% (*p* < 0.05), respectively. Intestinal short-chain fatty acids, especially butyric acid, were significantly increased (2.98 ± 0.72 µmol/mL, *p* < 0.05) and positively correlated with *Oscillospira* and *Collinsella*. Overall, ILA has the potential to regulate the gut microbiota, and an in-depth understanding of the relationship between tryptophan metabolites and gut microbiota is needed in the future.

## 1. Introduction

Tryptophan is an essential amino acid provided by dietary protein for human body. Based on daily intake, most of the protein (6–18 g/day) is digested and absorbed in the small intestine, while the rest reaches the colon and is degraded by the intestinal commensal microorganisms [1]. Tryptophan metabolites including indole ethanol (IE), indole acetic acid (IAA), indole acrylic acid (IA), indole-3-propionic acid (IPA), indole-3-lactic acid (ILA), indole-3-butyric acid (IBA) and indole-3-aldehyde (IAld) [1] can be generated by the gut microbiota such as *Bifidobacterium*, *Lactilactobacillus*, *Clostridium* and *Bacteroides* through the tryptophan metabolic pathway [2,3]. Nikolaus et al. found that the tryptophan levels in patients with inflammatory bowel disease (IBD) are significantly lower than those in healthy individuals [4]. The addition of tryptophan to the normal diet can alleviate the symptoms of colitis in mice induced by dextran sulfate sodium (DSS), improve intestinal permeability and reduce local inflammation [5,6]. Tryptophan metabolites are considered to be important signaling molecules for host–microbe interactions and may contribute to the homeostasis of the gut and even the whole body [7,8].

ILA is one of the important tryptophan metabolites, mainly produced by *Bifidobacterium* in probiotic conditioned media [9]. Relatively high ILA yields of 22.17–33.12 mg/L were detected in *Bifidobacterium infantis* using liquid chromatography-tandem mass spectrometry (LC-MS/MS) [9]. Recent studies have demonstrated that species of *Lactilactobacillus* such as *Ligilactobacillus*
*salivarius*, *Lactiplantibacillus*
*plantarum*, *L**imosilactobacillus*
*reuteri*, *Lacticaseibacillus*
*paracasei* and *Lactilactobacillus*
*sakei* can also produce ILA at levels of 4.30–30.70 mg/L [10]. *Lactilactobacillus* are symbiotic bacteria found in the gut of healthy humans, helping to regulate the intestinal microbiota and activate the body’s immunity [11]. Especially in newborns aged 1–3 days, the intestinal microbiota are structurally single and dominated by *Lactilactobacillus* [12]. Could these *Lactilactobacillus* from the gut in early life also metabolize ILA to promote gut health? 

The functional properties of ILA have been reported to mainly include free radical-scavenging activity, anti-inflammatory activity and immune regulation [13,14]. Walker et al. demonstrated that ILA metabolized by *B*. *infantis* in breast milk can suppress immune responses and promote the secretion of SCFAs by the intestinal microbiota, contributing to the prevention of necrotizing enterocolitis [15]. Chao et al. found that lemon exosome-like nanoparticles combined with *Lacticaseibacillus rhamnosus* GG (LGG, ATCC 53103) and *Streptococcus thermophilus* ST-21 can enhance the production of ILA, IAld, the active aryl hydrocarbon receptor (AhR) signaling pathway and enhance interleukin–22 (IL–22) secretion, leading to a decrease of *Clostridioides difficile* mortality in gut [16]. However, relevant studies on the regulation of gut microbiota by *Lactilactobacillus*-derived tryptophan metabolites are still limited [1]. Trillions of microorganisms in the human gut have a symbiotic relationship with the host and play important roles in nutrient supply, host defense and immunity in healthy individuals [17]. The dysbiosis of commensal microbiota in the host gut can easily trigger innate and adaptive immune response disorders, resulting in the development and prevalence of IBD, type 2 diabetes, and obesity [18]. The exploration of the regulatory effect of ILA on gut microbes will contribute to a comprehensive understanding of the impact of tryptophan metabolites on gut health.

In this study, we obtained high ILA-producing *Lactiplantibacillus plantarum* ZJ316 (CCTCC No. M 208077) screened from healthy newborn feces and purified ILA from the cell-free supernatant (CFS) by macroporous resin XAD-16, Sephadex G–25 and reverse-phase high-performance liquid chromatography (RP-HPLC). The effects of ILA on the regulation of the intestinal microbiota and SCFA metabolism were examined by 16S rRNA high-throughput sequencing and gas chromatography–mass spectrometry (GC-MS) analysis using an in vitro simulation model of human intestinal microbiota.

## 2. Materials and Methods

### 2.1. Determination of ILA Content

All wild lactic acid bacteria (LAB) strains were previously isolated from healthy newborn feces, fresh milk, cheese and yoghurt in our laboratory and identified by 16S rDNA sequencing in Sangon Biotech (Shanghai) Co., Ltd. (Shanghai, China) (Table 1) [19,20,21,22]. *L. rhamnosus* GG was purchased from American Type Culture Collection (ATCC). LAB strains were activated in Man Rogosa Sharpe Medium (MRS) at 37 °C, and two consecutive cultures (10^6^ CFU/mL inoculum) were incubated for 24 h. The CFS was obtained by centrifugation at 4 °C and 8000× *g* for 20 min. The content of ILA in the CFS of LAB was determined by analytical RP-HPLC (Waters, Milford, MA, USA) with a Waters SunFire C18 column (5 µm 4.6 × 250 mm, Waters, USA) and a UV detector (Waters e2695, USA). A linear elution gradient from 10% aqueous acetonitrile with 0.05% trifluoroacetic to 75% acetonitrile over 20 min was operated at a flow rate of 0.8 mL/min. The RP-HPLC peak areas of a series of ILA standards (1, 2.5, 5, 10, 25, 50, 100 mg/L) (Sigma-Aldrich, St. Louis, MO, USA) were measured with the injection volume of 30 µL. 

### 2.2. Purification of ILA from L. plantarum ZJ316

*L. plantarum* ZJ316 was selected as a high-producing strain, and fermented with 3% inoculation (*v*/*v*, 10^6^ CFU/mL inoculum) in 5 L MRS liquid medium (initial pH 6) for 24 h at 37 °C and 180 rpm in the lab-scale bioreactor Labfors 5 Bacteria (INFORS AG, Switzerland). The CFS of *L. plantarum* ZJ316 was adsorbed with macroporous resin XAD–16, and eluted by 2 L of ultrapure water, 30% methanol and 50% methanol (pH 7) at a flow of 1 mL/min, respectively. The 50% methanol eluent was collected, concentrated with a rotary evaporator (BUCHI R–215, Switzerland) and fractionated with Sephadex G–25 gel. Ultrapure water was used as the eluent solution at a flow rate of 1 mL/min. The eluents were collected every 3 min (3 mL/tube), and the ultraviolet-visible absorption was detected at 280 nm by a UV–2600 spectrophotometer (Shimadzu, Kyoto, Japan). The second fraction of Sephadex G–25 was further purified by RP-HPLC (Waters, USA) with a Waters SunFire C18 Prep column (5 µm 10 × 100 mm, Waters, USA) and UV-Vis detector (Waters 2998, USA). At a flow rate of 3 mL/min, a linear elution gradient from 10% aqueous acetonitrile with 0.05% trifluoroacetic to 40% acetonitrile was performed over 20 min, and the elution peaks were collected.

### 2.3. Identification of ILA

The molecular weight was identified by liquid chromatography-mass spectrometry (LC-MS) (Agilent 1200, Santa Clara, CA, USA). A total of 10 µL of purified sample was injected and detected by gradient elution: acetonitrile containing 0.1% formic acid (*v*/*v*) from 10% to 75% in 20 min, and then from 75% to 95% in the next 5 min at a flow rate of 0.5 mL/min and monitored at 280 nm. The mass spectrometric was quantified by an electrospray ionization source (ESI) in negative ion mode. The instrumentation settings were as follows: 350 °C of gas temperature, 9 L/min of gas flow, 45 psi of nebulizer pressure, 3500 V of capillary voltage, and 125 V of fragmentation voltage. 

### 2.4. Antibacterial Activity of ILA

The antibacterial activity of the ILA was examined using the Oxford cup method [23]. *Salmonella paratyphi-B* CMCC 50094, *Salmonella paratyphi-A* CMCC 50093, *Salmonella enterica* subsp. *arizonae* CMCC(B)47001, *Salmonella typhimurium* CMCC 50015 and *Escherichia coli* DH5α were purchased from the National Center for Medical Culture Collections (CMCC). *Salmonella enterica* subsp. *enterica* ATCC 14028, *Salmonella choleraesuis* ATCC 13312, *Pseudomonas aeruginosa* ATCC 47085 and *Listeria monocytogenes* ATCC 19,111 were purchased from American Type Culture Collection (ATCC). *Micrococcus luteus* CICC 10,209 was obtained from the China Center of Industrial Culture Collection (CICC). *Staphylococcus aureus* D48, *Staphylococcus warneri*, *Staphylococcus carnosus* pCA 44, *Staphylococcus carnosus* pot 20 and *Staphylococcus simulans* were gifted by Professor Eefjan Breukink, Utrecht University, The Netherlands. All strains stored in 20% (*v*/*v*) glycerol at –80 °C were grown in optimal medium (Luria-Bertani broth (LB), tryptone soya broth (TSB) or brain–heart infusion broth (BHI)) at 37 °C for 12 h, and propagated twice in the corresponding liquid broth at 180 rpm and 37 °C for 12 h. A 1% inoculum of bacterial suspension (10^6^ CFU/mL) at the exponential growth stage was mixed with a semi-solid medium and poured into plates pre-placed with Oxford cups. Then, 100 μL ILA (10 mg/mL) was added to each well after solidification. The diameter zones of inhibition were measured after 12–hour culture. ILA at final concentrations of 0, 0.1, 0.2, 0.4, 0.8, 1.6 and 3.2 mg/mL was added to 5 mL of liquid medium containing 1% indicator bacteria. The OD_600_ value of the bacterial suspension was determined to obtain the minimum inhibitory concentration (MIC) of ILA after culturing for 12 h [24].

### 2.5. Intestinal Simulation Model In Vitro

Eight healthy volunteers were selected according to following requirements [25]: 20–40 years old, BMI of 18–23, no gastroenteric disease, and no receipt of antibiotic treatment in the last three months. Two grams of fresh feces was collected each morning, dissolved with PBS (pH 7, 0.1 M), oscillated evenly, and filtered to a final concentration of 10% (*w*/*v*). YCFA medium was prepared according to the reported method [26]. All samples were randomly divided into a feces fermentation group (Con), a low-dose ILA group (ILA1), a medium-dose ILA group (ILA2) and a high-dose ILA group (ILA3). The ILA1, ILA2 and ILA3 groups were added with 5 mL YCFA medium and 500 μL fecal suspension with a final ILA concentration of 86 mg/L, 172 mg/L and 344 mg/L, respectively. The feces of the Con group were not treated before fermentation. After flushing with nitrogen to remove air, the fermentation flasks were sealed and incubated in an anaerobic workstation at 37 °C for 24 h.

### 2.6. 16S rRNA Sequencing

The DNA of the fermented fecal samples was extracted using a MagPure^®^ Soil DNA LQ Kit (Magen Biotechnology Co., Ltd., Guangzhou, China) according to the manufacturer’s instructions, and the DNA purity and concentration were checked by agarose gel and NanoDrop 2000 (Thermo Fisher, USA). The V3–V4 regions of 16S rRNA genes were amplified with the primers 343F (5’–TACGGRAGGCAGCAG–3’) and 798R (5’–AGGGTATCTAATCCT–3’). The PCR reactions were conducted using the following program: initial denaturation at 95 °C for 3 min; 35 cycles of denaturation at 95 °C for 30 s, annealing at 55 °C for 30 s, and elongation at 72 °C for 45 s; and a final extension at 72 °C for 10 min. The PCR products were recovered by a 2% agarose gel and purified using the AxyPrep DNA Gel Extraction Kit (Axygen Biosciences, Union City, CA, USA). The purified amplicons were sequenced with Illumina MiSeq platform (Biomarker Technologies Co., Ltd., Beijing, China). The paired-end reads of the raw sequencing data were quality-filtered by Trimmomatic and merged by FLASH. Operational taxonomic units (OTUs) were clustered with 97% similarity cutoff using UPARSE. The taxonomy of each 16S rRNA gene sequence was analyzed by the RDP Classifier algorithm compared with the Silva database (version 138) using a confidence threshold of 70%. The relative abundance of bacteria in the different groups was displayed by cumulative histogram, and the differences in bacterial community structure were estimated by α-diversity, β-diversity and linear discriminant analysis effect size (LEfSe).

### 2.7. SCFA Concentrations by GC-MS

One hundred microliters of fecal fermentation sample were thoroughly mixed with 50 μL phosphoric acid (15%), 10 μL caproic acid solution (75 μg/mL) and 140 μL ether for 1 min, and centrifuged at 12,000× *g* at 4 °C for 10 min to obtain the supernatant. A gas chromatography (GC) measurement was performed on a Trace 1300 gas chromatograph (Thermo Fisher Scientific, USA) with an Agilent HP-INNOWAX capillary column (0.25 μm 30 m × 0.25 mm, Agilent, USA). Helium was used as the carrier gas at 1 mL/min. Injection was made in split mode at 10:1 with a sample volume of 1 μL. The temperature of the injector, ion source and interface were 250 °C, 300 °C and 250 °C, respectively. The column temperature was increased from an initial temperature of 90 °C to 120 °C at 10 °C/min, and to 150 °C at 5 °C/min, and finally to 250 °C at 25 °C/min and maintained for 2 min. The mass spectrometric detection of metabolites was performed on an ISQ 7000 (Thermo Fisher Scientific, Waltham, MA, USA) using the single-ion monitoring mode with the electron energy of 70 eV.

### 2.8. Statistical Analysis

The data were presented as mean ± standard deviation. The graphs were displayed using GraphPad Prism 8.0.2 (GraphPad Software, San Diego, CA, USA). The differences between two groups were analyzed by a Student’s t-test. One-way analysis of variance (ANOVA) was used to compare the statistical differences among multiple groups. *p* values less than 0.05 were considered significant.

## 3. Results

### 3.1. Screening of High ILA-Producing LAB Strains and ILA Purification

The standard curve and regression equation of the ILA were obtained (Figure 1A), and the contents of ILA in the CFS of the LAB strains were calculated (Table 1). The strains in newborns feces had higher ILA production, especially *L. plantarum* ZJ316, with a yield of 43.14 ± 1.02 mg/L. The second was *L. paracasei* ZFM54 with a yield of 14.58 ± 0.25 mg/L, which was followed by *L. sakei* ZFM225 (6.39 ± 0.75 mg/L), *L. sakei* ZFM220 (12.92 ± 0.76 mg/L), *L. mesenteroide* ZFM802 (7.50 ± 0.02 mg/L) and *L. fermentum* ZFM001 (6.56 ± 0.35 mg/L). ILA was not detected in the CFS of the tested *L. rhamnosus*.

A total of 5 L of *L. plantarum* ZJ316 CFS was adsorbed and eluted by macroporous resin XAD–16. The 50% methanol eluent was collected and separated by a Sephadex G–25 gel column. As shown in Figure 1B, two peaks appeared on the “Tubes-Absorbance value” curve, and the second peak (G25–2) was further purified by RP-HPLC. The eluent “H4” at a retention time of 24.180 min was collected (Figure 1C). The eluent (H4) with a purity of 99.00% had a molecular weight of 205 Da (Figure 1D) and was identified as ILA (Figure 1E).

### 3.2. Antibacterial Activity and MICs of ILA

The results show that 100 µL of 10 mg/mL ILA treatment can significantly inhibit the growth of Gram-positive and Gram-negative bacteria (Table 2). The antibacterial diameters of ILA against six *Salmonella* strains ranged from 10.89 to 13.17 mm, among which *S. paratyphi-A* CMCC 50,093 was the most sensitive, with an MIC of 0.80 mg/mL. Compared with the effect on *Salmonella*, ILA had a stronger inhibitory activity on *Staphylococcus*, with an inhibitory diameter of 11.46–20.75 mm. ILA at 0.8 mg/L can inhibit most *Staphylococcus* species, among which *S. warneri* and *S. simulans* were the most sensitive, with an MIC of 0.40 mg/mL. The MICs of ILA against *E. coli*, *P. aeruginosa*, *M. luteus* and *L. monocytogenes* were 1.6, 3.2, 0.4 and 3.2 mg/mL, respectively.

### 3.3. Diversity Analysis of Bacterial Communities 

The effects of ILA intervention on the abundance and diversity of healthy gut microbiota were analyzed by 16S rRNA high-throughput sequencing. The shared and unique OTUs in different groups are shown in a Venn diagram (Figure 2A). The number of shared OTUs was 344. The Con group had the highest number of unique OTUs, while the unique OTUs in the low-, medium- and high-dose ILA intervention groups were 0, 1 and 0, respectively, indicating that ILA treatment had little effect on shaping the composition of the gut microbiota. Similarly, different concentrations of ILA also had no significant effect on bacterial alpha diversity compared with the Con group, as reflected in the abundance indexes (ACE and Chao1) and diversity indexes (Simpson and Shannon) (Figure 2B). Beta diversity was performed by principal coordinates analysis (PCoA) and partial least squares discrimination analysis (PLS-DA) based on unweighted UniFrac distance at the genus level (Figure 2C,D). There were significant differences between the Con group and the three ILA intervention groups. The sample points in the Con group were on the left side of the 0–point line, and the samples in the ILA intervention groups were on the right side. However, different concentrations of ILA treatment showed higher bacterial similarity. 

### 3.4. Relative Abundance of Bacterial Communities 

The top 13 phyla with the highest relative abundance in the fermented fecal samples are listed in Figure 3A. Compared with the Con group, ILA intervention showed greater effects on Proteobacteria, Bacteroidetes and Firmicutes. The relative abundances of Firmicutes and Bacteroidota in the ILA1, ILA2 and ILA3 groups were increased by 2.02%, 9.27% and 7.18%, and 9.61%, 15.38% and 8.13%, respectively. Proteobacteria decreased by 4.11%, 14.36% and 6.58%, respectively. ILA intervention also decreased the relative abundance of phyla Fusobacteria and Verrucomicrobia. No significant changes were found in the relative abundance of bacteria in Actinobacteria, as well as the Firmicutes/Bacteroidota ratio. 

At the genus level, the top 25 species in relative abundance are shown in Figure 3B. Compared with the Con group, the average relative abundance of fecal microbiota was more affected by the medium-dose ILA (ILA2) than by the low- and high-dose ILA (ILA1 and ILA3) intervention. Four genera with significant differences were selected: *Bifidobacterium*, *Faecalibacterium*, *Escherichia* and *Phascolarctobacterium* (Figure 3C). In the ILA2 group, the average relative abundance of *Bifidobacterium* and *Faecalibacterium* increased significantly from 3.69 ± 1.24% to 5.36 ± 2.31% and from 1.21 ± 0.45% to 2.19 ± 0.77% (*p* < 0.01), respectively. However, the abundance of *Escherichia* and *Phascolarctobacterium* decreased from 23.81 ± 6.24% to 16.41 ± 4.81% (*p* < 0.05) and from 4.35 ± 1.45% to 2.84 ± 1.02% (*p* < 0.05), respectively. In addition, medium-dose ILA also promoted the growth of *Dislister* and *Dore* in the feces in vitro. 

### 3.5. Communities Difference by LEfSe Analysis

The effect sizes of each differentially abundant bacterial taxon were assessed by LEfSe (LDA > 3.0, *p* < 0.05). In Figure 4A, *Ilumatobacter*, *Muribaculum*, *Lactilactobacillus*, *Lactococcus* and *Streptococcus* were the potential biomarkers in the Con group, while species of *Bacteroides coprocola*, *Desulfovibrio fairfieldensis* and *Parasutterella excrementihominis* were common in the ILA2 group. In contrast, the bacterium with the most significant abundance difference in the ILA3 group was *Phascolarctobacterium succinatutens*. The histograms in Figure 4B show that the bacterial abundance of the genera *Lactilactobacillus*, *Muribaculum* and *Ilumatobacter* were all significantly decreased in the ILA-treated samples.

### 3.6. Correlation of SCFA Metabolism and Gut Microbiota

The SCFA contents were quantitatively analyzed by GC-MS after 24 h anaerobic fermentation in vitro (Figure 5A–D). The contents of the total acid, acetic acid, propionic acid and butyric acid of normal feces were 7.43 ± 1.52 μmol/g, 2.48 ± 0.63 μmol/g, 2.03 ± 0.71 μmol/g and 1.93 ± 0.65 μmol/g, respectively, and tended to increase after ILA intervention, especially in the medium-dose group. The total acid content increased to 9.04 ± 2.00 μmol/g, acetic acid to 2.86 ± 0.52 μmol/g, propionic acid to 2.26 ± 0.32 μmol/g and butyric acid to 2.98 ± 0.72 μmol/mL (*p* < 0.05). 

In addition, the combined evaluation of the SCFA contents and bacterial abundance at the genus level was analyzed by Pearson’s correlation coefficient (Figure 5E). The red in the correlation matrix diagram is a positive correlation, and the blue is a negative correlation; the darker the color, the greater the correlation. It was shown that total acid was positively associated with the relative abundance of *Oscillospira* (*p* < 0.05) and *Prevotella* (*p* < 0.05), and negatively associated with *Bacteroides* (*p* < 0.05) and *Roseburia*. Propionic acid was mainly associated with *Allisonella* (*p* < 0.05). Butyric acid was related to the abundance of its producers such as *Oscillospira* and *Collinsella*.

## 4. Discussion

Gut microbiota-derived tryptophan metabolites play important roles in hosts’ homeostasis [27]. Indole-3-lactic acid has intestinal probiotic functions such as antioxidant activity [10], immune regulation [2] and inflammation reduction [28]. ILA can be produced by LAB through tryptophan metabolism, especially infant-type human-residential *Bifidobacteria* such as *B. longum subsp. longum*, *B. longum subsp. infantis*, *B. breve* and *B. bifidum* [9]. The ability to produce ILA also reflected strain-specific features. The highest ILA yield of *B. longum subsp. infantis* ATCC15697 was 33.12 μg/mL [29]. *L. plantarum* F51 and *L. plantarum* UM55 can also produce ILA, with yields of 3.63 mg/L and 6.70 mg/L, respectively [10,30]. In this investigation, we tested the ILA-producing ability of various LAB strains screened from newborn feces and fermentation by RP-HPLC. We found that the ILA content in the CFS of *L. plantarum* ZJ316 was 43.14 ± 1.02 mg/L, the highest yield among all strains. ILA with a purity of 99.00% was firstly purified via macroporous resin, Sephadex and RP-HPLC. Compared with the commonly used ultrafiltration centrifugation [29], this method is beneficial to distinguish substances with similar molecular weights and to improve ILA purity.

Tryptophan metabolites were well described as an intercellular signal molecule that affects spore formation, plasmid stability, drug resistance, biofilm formation and virulence [31]. ILA has previously been reported to have antibacterial activity against *Penicillium* sp. [32], *E**. coli* and *Bacillus cereus* [33]. Here, we demonstrated that ILA had antibacterial activity against foodborne pathogens such as *S. enterica* Typhimurium and *S. aureus* that caused intestinal microbial disturbance and tissue inflammation [34]. It appears that ILA has the potential to act as a regulator of gut microbiota by inhibiting the growth of harmful bacteria. The important associations between tryptophan metabolites and gut microbial species were further identified by establishing a simulated intestinal flora model in vitro. 

Gut microbiota analysis revealed that ILA can significantly stimulate the growth of many bacteria known to promote human health, including the genera *Bifidobacterium*, *Faecalibacterium, Dislister* and *Dore*. *Bifidobacterium* are beneficial microbiota of the gut and have many important physiological functions such as tumor suppression [35], immune enhancement [36], allergy relief [37], and inflammation reduction [38]. Laursen et al. reported that breastfeeding significantly increased the abundance of *Bifidobacteria* in the feces of healthy infants and improved early-life immune function through metabolically produced ILA [2]. Ehrlich et al. also found that *B. infantis* predominated in the stool of healthy breastfed infants, increased the content of ILA, and decreased IL–8 in TNF-α and LPS-induced macrophage and intestinal epithelial cells [39]. It may be speculated that there is a mutually beneficial association between intestinal ILA and *Bifidobacterium*. The genus *Faecalibacterium* is one of the most prevalent species in the gut microbiome of healthy human adults, and *F. prausnitzii* is the only known species [40]. Previous studies suggested that a decreased abundance of *Faecalibacterium* is associated with inflammatory bowel disease (IBD) as well as colorectal cancer and diabetes [41]. *F. prausnitzii* is currently one of the most promising taxa for developing next-generation probiotics [42]. In addition, the intervention of ILA reduced the relative abundances of *Escherichia*, *Phascolarctobacterium*, *Muribaculum* and *Ilumatobacter*. It has been reported that *Phascolarctobacterium*, the producer of intestinal acetate and propionate, may be related to the host’s metabolism and mood, and decreases as people continue to age [43]. However, what is still not clear is the mechanism by which the relative abundance of *Lactilactobacillus* decreased after ILA intervention. Similarly, IE has been identified as a quorum-sensing factor in fungi [44], and exerts antimicrobial activity against *S. aureus*, *S. enterica* and *L. plantarum* [45].

Changes in gut commensal bacteria also affect the SCFA metabolism. Acetic acid, propionic acid, and butyric acid together account for approximately 90–95% of the total SCFA [46]. SCFA is an important effector of gut microbiota and is not only involved in energy metabolism, but also has potential applications in the prevention and treatment of IBD, obesity and diabetes [47]. Butyric acid, in particular, is known to be an important indicator of gut health [48]. ILA intervention contributed to the metabolism of intestinal SCFAs, leading to an increase in total acid, acetic acid, propionic acid and butyric acid. Butyric acid may be related to the increased abundance of *Faecalibacterium* and *Bifidobacterium*. Ehrlich et al. found that *B. longum subsp. infantis* were primarily in the gut of breast-fed infants, with significantly elevated levels of the tryptophan metabolites ILA, acetic and lactic acid in feces [39]. The relationship between tryptophan metabolites and SCFA metabolism in the gut microbiota needs to be further studied.

## 5. Conclusions

In summary, we identified that ILA produced by *Lactilactobacillus* strains from the feces of healthy newborns has the potential to modulate the gut microbiota, especially in promoting the growth of probiotics such as *Bifidobacterium* and *Faecalibacterium*, and inhibiting *Escherichia* and *Phascolarctobacterium*. Meanwhile, ILA affected the SCFA metabolism of gut microbiota and significantly increased the butyric acid production. Further study of how *Lactilactobacillus* contribute to gut microbiota homeostasis by producing ILA could also identify the gut host–microbe crosstalk in health and disease.

## Figures and Tables

**Figure 1 foods-11-03302-f001:**
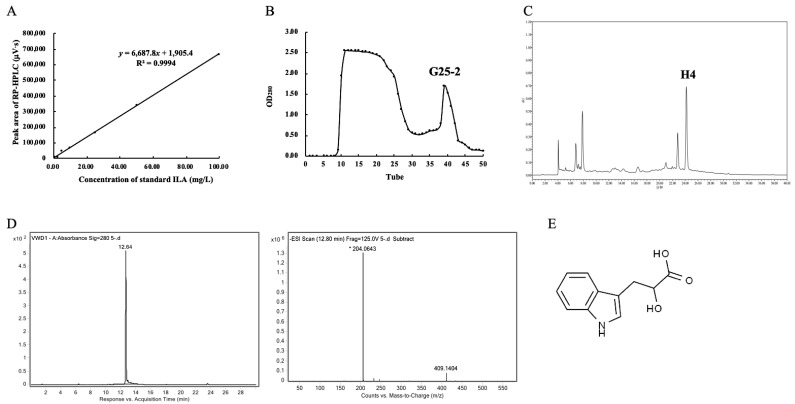
Screening of high ILA-producing strains and purification of ILA from *L. plantarum* ZJ316. (**A**) Standard curve of ILA by RP-HPLC. (**B**) Separation of the 50% methanol eluent with Sephadex G–25 column. (**C**) Purification of the fraction G25–2 by RP-HPLC. Elution “H4” was collected. (**D**) Identification of ILA by LC-MS. (**E**) Molecular structure of ILA.

**Figure 2 foods-11-03302-f002:**
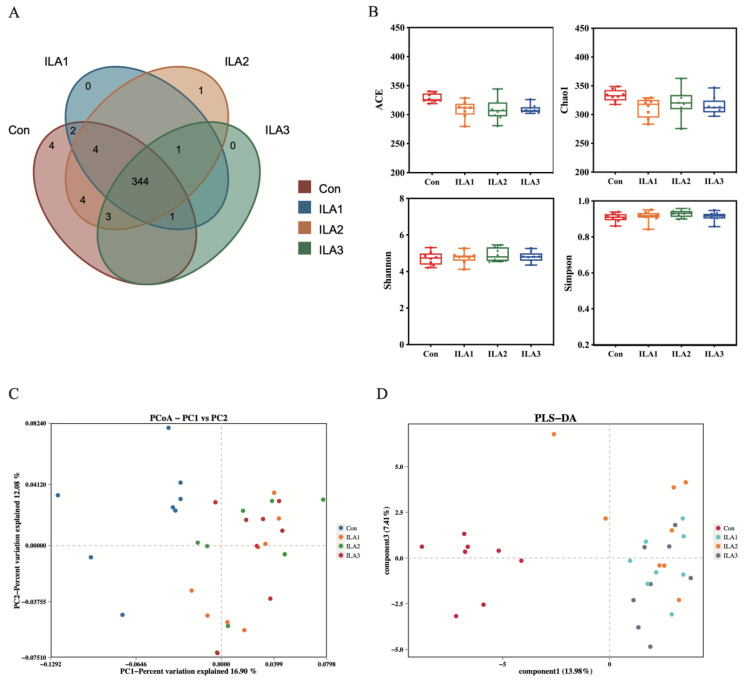
Alpha diversity and beta diversity analysis of bacterial communities in fecal fermentation samples. (**A**) Venn diagram of microbial compositions on OTU level. (**B**) Alpha diversity including the ACE, Chao1, Shannon and Simpson indexes based on OTU level. There were no significant differences in the microbial richness and diversity between the Con and ILA groups (*p* > 0.05). (**C**) PCoA plots at genus level based on unweighted UniFrac analysis. (**D**) PLS-DA plots at genus level based on unweighted UniFrac analysis.

**Figure 3 foods-11-03302-f003:**
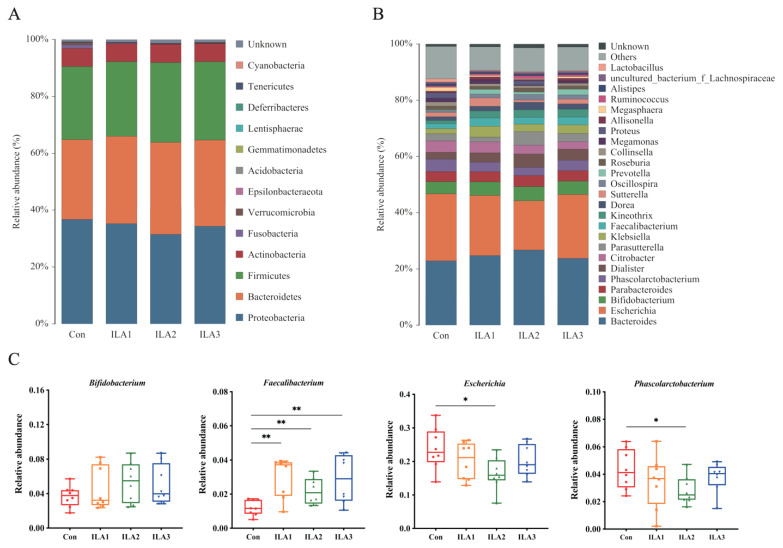
Changes in the relative abundances of bacterial communities. (**A**) Distribution of the predominant bacteria at the phylum level. (**B**) Distribution of the predominant bacteria at the genus level. (**C**) Compositional alterations of genera *Bifidobacterium*, *Faecalibacterium*, *Escherichia* and *Phascolarctobacterium*. Differences between Con group and each ILA group were analyzed by Student’s *t*-test. *, *p* < 0.05; **, *p* < 0.01.

**Figure 4 foods-11-03302-f004:**
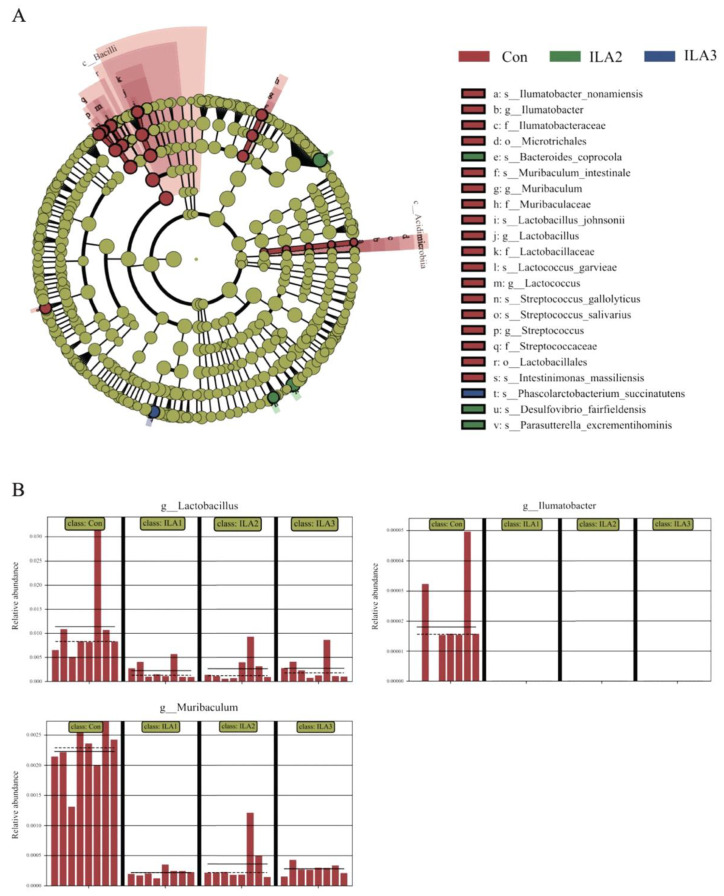
Differentially abundant taxa of gut microbiota by LEfSe analysis. (**A**) Most abundant taxa from the phylum to the species level in fermented feces samples. Only taxa with an LDA score greater than 3.0 are shown. (**B**) Relative abundance of *Lactilactobacillus*, *Muribaculum* and *Ilumatobacter* in each sample.

**Figure 5 foods-11-03302-f005:**
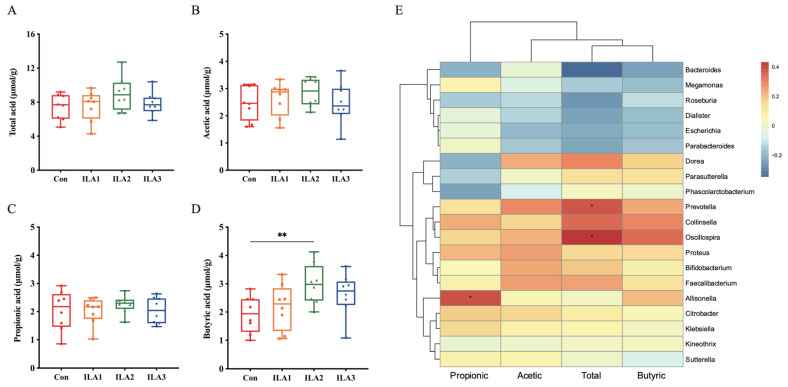
The content of SCFAs in fecal samples and the correlation analysis between SCFAs and microbiota. (**A**–**D**) Changes of SCFA contents included total acid, acetic acid, propionic acid and butyric acid in fecal samples. Differences between Con group and ILA group were analyzed by Student’s *t*-test. **, *p* < 0.01. (**E**) Heatmap of the relative abundance of gut microbiota and main SCFAs by Pearson. *, *p* < 0.05.

**Table 1 foods-11-03302-t001:** Screening of ILA-producing lactic acid bacteria strains.

Lactic Acid Bacteria	Sources	Culture Condition	Retention Time (min)	ILA Content (mg/L)
*Lactiplantibacillus plantarum* ZFM9	Healthy newborn feces	37 °C, MRS	22.720	25.08 ± 1.13
*Lactiplantibacillus plantarum* ZJ316		37 °C, MRS	22.765	43.14 ± 1.02
*Lactiplantibacillus plantarum* ZFM55		37 °C, MRS	22.734	30.89 ± 0.57
*Lactiplantibacillus plantarum* LZ206		37 °C, MRS	22.699	22.31 ± 0.81
*Lacticaseibacillus paracasei* ZFM54		37 °C, MRS	22.743	14.58 ± 0.25
*Lactiplantibacillus plantarum* LZ227	Fresh milk	37 °C, MRS	22.697	12.55 ± 0.24
*Lactilactobacillus sakei* ZFM225		37 °C, MRS	22.700	6.39 ± 0.75
*Lactilactobacillus sakei* ZFM220		37 °C, MRS	22.727	12.92 ± 0.76
*Lacticaseibacillus rhamnosus* ZFM231		37 °C, MRS	/	/
*Lacticaseibacillus rhamnosus* ZFM202		37 °C, MRS	/	/
*Lactiplantibacillus plantarum* ZFM806	Cheese	37 °C, MRS	22.719	36.35 ± 1.37
*Leuconostoc mesenteroides* ZFM802		37 °C, MRS	22.712	7.50 ± 0.02
*Limosilactobacillus fermentum* ZFM001	Yogurt	37 °C, MRS	22.685	6.56 ± 0.35
*Lacticaseibacillus rhamnosus* GG	Commercial strain	37 °C, MRS	/	/

Note: MRS, Man Rogosa Sharpe Medium; “/”, not detected.

**Table 2 foods-11-03302-t002:** Antibacterial activities and MICs of ILA.

	Indicator Bacteria	Sources	Culture Condition	Inhibition Zone Diameter (mm)	MIC (mg/mL)
G^−^	*Salmonella paratyphi*-B CMCC 50094	CMCC	37 °C, LB	11.24 ± 0.35	1.6
	*Salmonella paratyphi*-A CMCC 50093	CMCC	37 °C, LB	12.89 ± 0.28	0.8
	*Salmonella enterica* subsp. *arizonae* CMCC(B)47001	CMCC	37 °C, LB	11.36 ± 0.26	1.6
	*Salmonella enterica* subsp. *enterica* ATCC 14028	ATCC	37 °C, LB	12.35 ± 0.44	1.6
	*Salmonella choleraesuis* ATCC 13312	ATCC	37 °C, LB	11.51 ± 0.34	1.6
	*Salmonella typhimurium* CMCC 50015	CMCC	37 °C, LB	12.28 ± 0.40	1.6
	*Escherichia coli* DH5α	CMCC	37 °C, LB	12.50 ± 0.17	1.6
	*Pseudomonas aeruginosa* ATCC 47085	ATCC	37 °C, LB	10.66 ± 0.08	3.2
G^+^	*Staphylococcus aureus* D48	Gift from Eefjan Breukink, Utrecht University, The Netherlands	37 °C, LB	11.93 ± 0.47	0.8
	*Staphylococcus warneri*		37 °C, LB	20.25 ± 0.50	0.4
	*Staphylococcus carnosus* pCA 44		37 °C, LB	12.93 ± 0.40	1.6
	*Staphylococcus carnosus* pot 20		37 °C, LB	14.73 ± 0.53	0.8
	*Staphylococcus simulans*		37 °C, LB	15.13 ± 1.15	0.4
	*Micrococcus luteus* CICC 10209	CICC	30 °C, TSB	19.06 ± 0.76	0.4
	*Listeria monocytogenes* ATCC 19111	ATCC	37 °C, BHI	10.95 ± 0.07	3.2

Note: CMCC, National Center for Medical Culture Collections; ATCC, American Type Culture Collection; CICC, China Center of Industrial Culture Collection; LB, Luria–Bertani broth; TSB, tryptone soya broth; BHI, brain–heart infusion broth.

## Data Availability

The data used and analyzed during the current study are available from the corresponding author on academic request (Q.G.). The data are not publicly available to preserve the privacy of the data.

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
