# Peer review of "The Effect of Indole-3-Lactic Acid from Lactiplantibacillus plantarum ZJ316 on Human Intestinal Microbiota In Vitro"

_foods, 2022, doi:10.3390/foods11203302_

Round 1
Reviewer 1 Report
Comment for authors:
№ |
Line number |
Comment |
1 |
16 |
“Staphylococci spp.” incorrect genus name |
2 |
16 |
“in-vitro” misspelled |
3 |
22 |
“SCFAs” - decipher the abbreviation |
4 |
41 |
“IBD” - decipher the abbreviation |
5 |
51-52 |
“Lactobacillus spp. such as L. salivarius, L. plantarum, L. reuteri, L. paracasei and L. sakei” – all lactobacilli species should be decipher while mentioning them first time |
6 |
61-64 |
“Walker et al. confirmed that ILA metabolized by B. infantis in breast milk can suppress immune response and secret short-chain fatty acids (SCFAs), contributing to prevent necrotizing enterocolitis (NEC)” - It is not entirely clear - can ILA secret SCFAs? or B. infantis are secreting SCFAs? |
7 |
65 |
“L. rhamnosus GG” - should be decipher while mentioning them first time |
8 |
66 |
“AhR” - decipher the abbreviation |
9 |
66 |
“IL-22” - decipher the abbreviation |
10 |
77 |
“CFS” - decipher the abbreviation |
11 |
78 |
“RP-HPLC” - decipher the abbreviation |
12 |
81 |
“in-vitro” misspelled |
13 |
84 |
“LAB” - decipher the abbreviation |
14 |
84-85 |
“All wild LAB strains were isolated from healthy infant feces and fresh milk previously” – References needed |
15 |
86 |
“MRS” - decipher the abbreviation |
16 |
122 |
“The diameters of antibacterial zone” – maybe better the diameters zone of inhibition? |
17 |
128-130 |
“Eight healthy volunteers were selected according to following requirements: 20 ~ 40 years old, 18-23 of BMI, no gastroenteric disease, no receiving antibiotic treatment in three months” - How you can know that those volunteers do not have other diseases for example type 2 diabetes, CVD, and others noncommunicable diseases? |
18 |
140 |
“DNA of fecal fermentation samples was extracted using MagPure Soil DNA LQ Kit (Magen Biotechnology Co., Ltd., Guangzhou, China)” – Why extraction of DNA from fecal fermentation samples was performed by using MagPure Soil DNA LQ Kits which are specifically designed for Soil DNA extraction? |
19 |
156 |
“GC measurement” - decipher the abbreviation |
20 |
181 |
By which methods all LAB strains was identified? Why you did not take for screening bifidobacteria strains? |
21 |
197 |
“Staphylococci spp.” incorrect genus name |
22 |
204 |
“Staphylococci” incorrect genus name |
23 |
204 |
“Antibacterial diameter” – maybe “Inhibition zone diameter”? |
24 |
204 |
On the basis of which parameters indicator strains were selected? Why only two genera were selected? |
Reviewer 2 Report
Dear Editor and Authors,
Major comments
1-How other inhibitors produced by lactic acid bacteria such as bacteriocins and organic acids were removed?
2-Why were these species of bacteria chosen as an indicator of the inhibition method? It was better for you to use other species that are more present in the intestine, such as fecal coliform bacteria (E.coli).
Minor comments
1-The modern nomenclature of lactic acid bacteria should be used throughout the manuscript such as Lactobacillus plantarum to Lactiplantibacillus plantarum.
2-Page 3 line 86, line 95, How many microorganisms are in the inoculation size?
3- All working methods mentioned in the manuscript do not contain scientific references, Why????
I suggest some references of this methods
Page 3 line 87, Niamah, A. (2019). Ultrasound treatment (low frequency) effects on probiotic bacteria growth in fermented milk. Future of Food: Journal on Food, Agriculture and Society, 7(2), Nr-103.
Page 3 line 93, Szkop, M., & Bielawski, W. (2013). A simple method for simultaneous RP-HPLC determination of indolic compounds related to bacterial biosynthesis of indole-3-acetic acid. Antonie Van Leeuwenhoek, 103(3), 683-691.
Page
4- The antibacterial activity is unclear.
What are the bacterial isolates that were used as an indicator?
How are these bacteria activated? What are the sources of these bacteria?
The size of the bacterial inoculator here is very important? How many viable cells are in this size?
Add referenceto this method (Al-sahlany, S. T. G., Altemimi, A. B., Abd Al-Manhel, A. J., Niamah, A. K., Lakhssassi, N., & Ibrahim, S. A. (2020). Purification of Bioactive Peptide with Antimicrobial Properties Produced by Saccharomyces cerevisiae. Foods, 9(3).
5-16S rRNA sequencing method, What is the program of PCR??
6-Scientific names of bacteria must be written correctly throughout the manuscript.
Some names contain errors such as Staphylococci correct to Staphylococcus, see page 1 line 16 , Table2 and.........etc.
7- In page 3 line 84, (All wild LAB strains were isolated from healthy infant feces and fresh milk) These bacterial isolates how isolated and how diagnosed? What medium is it grown?
Sources of isolation are milk and baby faeces. As for the results, Table 1 shows cheese and yoghurt?
8-The conclusions in the manuscript need to be rewritten again because there are some results within the conclusions.

Round 2
Reviewer 2 Report
Dear editor(s),
The authors made all the necessary changes to improve the manuscript, and now I recommend it for publication in its current form.